# Pharmacists as a Source of Advice on Medication Use for Athletes

**DOI:** 10.3390/pharmacy8010010

**Published:** 2020-01-15

**Authors:** Kwang Choon Yee, Michael De Marco, Mohammed S. Salahudeen, Gregory M. Peterson, Jackson Thomas, Mark Naunton, Sam Kosari

**Affiliations:** 1Discipline of Pharmacy, Faculty of Health, University of Canberra, Bruce, ACT 2614, Australia; Kwang.Yee@canberra.edu.au (K.C.Y.); m.demarco@live.com.au (M.D.M.); g.peterson@utas.edu.au (G.M.P.); jackson.thomas@canberra.edu.au (J.T.); Mark.Naunton@canberra.edu.au (M.N.); 2School of Pharmacy and Pharmacology, University of Tasmania, Hobart, TAS 7005, Australia; mohammed.salahudeen@utas.edu.au

**Keywords:** drugs, sport, athletes, pharmacists, advice, doping, medication

## Abstract

**Background**: The World Anti-Doping Agency (WADA) specifies substances that competitive sportspersons are not allowed to take. Some of these substances are contained in common medicines used in everyday medical practice and could be used by athletes by accident. **Objectives**: This study aimed to explore pharmacists’ knowledge and confidence in guiding athletes about the use of medicines in professional sport. **Methods**: Registered pharmacists in Australia were invited to participate in an online survey. The survey had five domains and aimed to identify pharmacists’ demographic information, interest in sport, familiarity with WADA guidelines, knowledge on prohibited drug classes, and their opinion about the role of pharmacists in educating athletes on medication use. Descriptive statistics were provided and where appropriate, Chi-square, Mann–Whitney and independent t-test were used to identify potential associations and difference between means. **Results**: One hundred and thirty-five pharmacists (response rate of 10.6%) completed the survey, with the majority indicating that they were not confident in advising athletes on medication use. Although most respondents believed that pharmacists have a role in the education of athletes to help avoid unintentional doping, only about a quarter indicated that they had sufficient knowledge to advise athletes. About one-half of the respondents could provide fully correct answers when asked to identify the WADA status of some commonly used drugs. **Conclusions**: The results of the survey indicate that upskilling is required to enable pharmacists in Australia to provide accurate medication advice to professional athletes.

## 1. Introduction

Medication use among athletes is very common [1]. Often they are taken for medical illness or injury, but there is the potential for medication abuse [2] in order to gain a competitive edge [3]. While some medications may provide benefit to an athlete in improving their fitness and/or performance, they are intended to be used under the supervision of health professionals for the treatment of specific illnesses. When the medication is being misused to gain an unfair competitive advantage, it may also produce significant negative consequences to the person’s health [2]. Therefore, “performance-enhancing drugs” are banned for use by athletes under the prohibited list published by the World Anti-Doping Agency (WADA) [4]. The WADA list is updated every year. 

A substance or a drug (or class of drugs without naming all agents in the class) can be included in the WADA prohibited list for various reasons [4,5]. For example, anabolic steroids can increase an athlete’s physical strength by increasing muscle mass, but may produce long-lasting adverse effects, such as psychoses, hepatitis, acceleration of cancer growth, diabetes mellitus, dyslipidaemia, cardiomyopathy, and nephrotoxicity [4]. Some drugs, such as diuretics, are included in the WADA prohibited list not because they can improve athletic performance, but because they can conceal the use of performance-enhancing drugs [4]. In addition, some drugs are prohibited only during the competition (e.g., narcotics), and some drugs are only prohibited in specific sports (e.g., beta-blockers in golf) [4]. 

Athletes are permitted to use a specified drug on the WADA list for a legitimate medical reason under the Therapeutic Use Exemptions (TUE) rule (e.g., use of a beta-2-agonist for the treatment of asthma) [4]. However, there is often a maximum dose that an athlete can use which is reasonable for the therapeutic benefit and should be approved by the local authority (e.g., Therapeutic Goods Administration for Australia, or the Food and Drug Administration for the USA), whereas a significantly higher dose can indicate its use for a “performance-enhancing” purpose [4,5]. Additionally, WADA states that prohibited drug groups will include all drugs that exhibit specific mechanisms of action including future substances and substances not individually named [4].

According to WADA, there are more than 660 sport organisations accepted the World Anti-Doping Code in 2019, including the Australian Sports Anti-Doping Authority (ASADA). According to the WADA code, all athletes (elite or non-elite) under the signatories will need to comply with the prohibited list. For example, ASADA stated that any athlete who competes in an international level competition, domestic (national level) competition, or “satisfies the definition of an athlete under the National Anti-Doping Scheme” is subject to anti-doping testing [5]. WADA states that an athlete has to bear the responsibility of any drug use, regardless of who has provided or recommend the use of such substance [4]. The majority of elite athletes would have access to assistance from specialist trainers/doctors, and the athletes may take supplements or medications based on advice without knowing what they contain [6,7,8]. However, “accidental doping” still occurs due to the decision made by specialist trainers/doctors. One example occurred at the 2000 Summer Olympic Games in Sydney, where an athlete was accused of taking a prohibited medication for common cold, that was given by the team physician [9]. Another notable example is the use of medlonium by Maria Sharapova in 2016 [10]. A list of doping violation cases, intentional or accidental, are available via sport organisations such as ASADA, Federation Internationale De Nation, Athletics Integrity Unit.

On the other hand, non-elite athletes will typically need to depend on the advice provided by their regular healthcare providers, such as the general practitioner (GP) or pharmacist, to avoid the unintentional use of prohibited substances [6,7,11,12]. Therefore, it is imperative for the pharmacist to provide appropriate medication advice to athletes [6]. However, there is currently no formally defined role for pharmacists in this process [13]. A recent systematic review reported that although pharmacists showed a willingness and positive attitude to counsel athletes, however, limited relevant education and lack of confidence have prevented them from having a significant role in this area [13]. 

This study aimed to assess the knowledge of pharmacists on performance-enhancing medications and their opinion about the role of pharmacists in providing advice to professional athletes on medication use.

## 2. Materials and Methods

An electronic survey questionnaire was developed using SurveyMonkey. This is an original survey created based on information gathered via literature and consultation with experts in the field. The survey was face validated by three reviewers including academics and practising pharmacists with prior experience of research in sports medicine, survey design, and practising pharmacy in various settings to check for appropriateness and relevance of questions. The survey was sent to 1277 pharmacists in Australia (accounting for 4% of the total 29,123 registered pharmacists in Australia) [14] by email, from October 2016 to December 2016. Using convenience sampling, pharmacists’ email addresses were identified through publicly available records in which pharmacists who were members of the organisations agreed to publish their contact detail, including their email address. Those organisations included the Australian Association of Consultant Pharmacists, community pharmacy groups and pharmacy websites. Up to two reminder emails were sent to the contact list. The short survey consisted of 7 questions to identify: pharmacists’ demographic information, interest in sport, familiarity with WADA guidelines, knowledge on prohibited drug classes, and their opinion about the role of pharmacists in educating athletes on medication use. Statistical analyses were conducted with SPSS version 23 (IBM, Armonk, NY, USA). Descriptive statistics were conducted and where appropriate, Chi-square and Mann–Whitney was used to identify potential associations among categories (parametric and non-parametric data respectively), and the difference between means (parametric data only) were determined by independent t-tests. The project was approved by the University of Canberra Human Research Ethics Committee (16-134).

## 3. Results 

The study included 135 pharmacists who completed the survey, with a response rate of 10.6%. Community pharmacists represented the majority of the respondents, and a larger variety of pharmacists represented the other roles, including pharmacists who perform home and residential aged care medication reviews, work in GP clinics, academia, pharmaceutical industry, as well as pharmacists working at multiple positions. Among the respondents, about one-half (n = 69) indicated that they were aware of where to find the WADA prohibited list. Not surprisingly, pharmacists who had an interest in sports (determined by the frequency they watch sport on TV) were more likely to be aware of the WADA prohibited list (*p* = 0.01; 2-tailed chi-square test). Although most respondents believed that pharmacists have a role in the education of athletes to help avoid unintentional doping, only about a quarter of them (n = 39) indicated that they had sufficient knowledge to advise athletes. This lack of knowledge was supported by the result of the survey question that assessed the respondents’ knowledge on identifying drugs that are prohibited in professional sports; only half of the respondents (n = 69) could provide fully correct answers. The responses to the survey questions are summarised in Table 1.

Upon further analysis, we identified that pharmacists who were aware of the WADA prohibited list were more likely to correctly identify the prohibited medications (*p* = 0.02; 2-tailed chi-square test). Other demographic factors were not found to have any significant influence, including the nature of the workplace or the pharmacists’ interest in sports. We did not observe any significant relationship among other parameters.

## 4. Discussion

This study reports of limited pharmacists’ knowledge and confidence in providing advice to athletes about medication use. In healthcare, identification, treatment, and prevention of significant illness are the primary focus for most professions. The nature of prohibited medication use in sport is often viewed as a small subsection from the mainstream of healthcare. Pharmacists, as well as many other healthcare professionals, seem to have some general knowledge about the nature of the prohibited drug list, but few are willing and/or able to provide expert service [13], as identified in our study. 

Findings of this Australian study are consistent with the reports of international studies in which pharmacists self-reported that their knowledge and awareness of doping, anti-doping and sport supplements are low [13,15,16]. Additionally, pharmacists reported low confidence [15] and difficulty finding reliable, evidence-based information about drugs in sport and sport supplements [17]. Another study in Qatar showed that 60% of pharmacist respondents were unaware of WADA’s role in doping. Overall, there is a small number of studies in the international literature that investigated the role of pharmacists in anti-doping and sports medicine [12].

Furthermore, the majority of athletes and trainers have limited knowledge about medications, supplements, and chemicals that they use [3,7]. Many of them depend on the advice of primary healthcare providers, such as GPs or community pharmacists, to avoid unintentional doping [8,18]. However, the results of our survey suggest that some upskilling of pharmacists will be required to avoid misguided trust. Recent programs that have been used for expanding pharmacist’s services in other areas, such as asthma educator training and medication review accreditation, have been successful in Australia and may serve as a model. However, the most appropriate method of upskilling and the role of upskilled pharmacists will need to be determined by future studies. 

This scoping study provides some useful information about pharmacists’ knowledge in regard to the WADA list of performance-enhancing medications, and it can provide a useful base for further studies. 

Limitations of this study include small sample size and a relatively low response rate. The questionnaires are focused on a relatively narrow field of interest and contain only a small number of questions to test the knowledge of pharmacists, which was designed with the aim of encouraging busy pharmacists to participate in this survey. The majority of responders were based in community pharmacy; however, this may not be representative of the entire pharmacist community in Australia.

## 5. Conclusions

Pharmacists can potentially provide specialised advice to athletes because of their advanced knowledge in drugs and being health professionals who are easily accessible to the public [8,11,12,18,19]. This is particularly important, given the new WADA specifications that prohibited drug groups will include all drugs that exhibit specific mechanisms of action (including future substances and substances not individually named) [4]. However, this study indicates that upskilling is required to enable pharmacists to accurately and confidently provide medication-related advice to athletes, with the aim of preventing unintentional doping. Upskilling can be achieved by specific accreditation or through workshops, short courses, and/or continuing professional development activities [13]. An example of a similar initiative was accomplished by the International Olympic Committee (IOC), who introduced a new certificate for sports pharmacists to learn about effective clinical drug use and doping prevention in athletes [20]. However, the options will need to be investigated and validated in the future.

## Figures and Tables

**Table 1 pharmacy-08-00010-t001:** Survey responses (n = 135).

Survey Question	Response Rate
Where is your primary workplace as a registered pharmacist?	Community 62% Hospital 13% Others 25%
In the past 12 months, how often did you watch the professional sport (on TV or live)?	≥Once a week 39%Once a fortnight 12%Once a month 10%<Once a month 39%
Are you aware that professional athletes have been suspended from the competition as a result of using performance-enhancing medications unintentionally?	Yes (97%)No (3%)
Are you aware of where to find the WADA guideline on prohibited performance-enhancing medications?	Yes (59%)No (41%)
Which of the following drug(s) is banned in professional sport?	
● Anabolic steroids	Correct answer (99%)
● Beta-2 agonists	Correct answer (69%)
● NSAIDs	Correct answer (90%)
● Diuretics	Correct answer (84%)
● PPIs	Correct answer (98%)
● Stimulants (e.g., methylphenidate)	Correct answer (96%)
● Paracetamol	Correct answer (99%)
Do you believe you have sufficient knowledge to treat and educate athletes on performance-enhancing medications?	Yes (29%)No (54%)Unsure (17%)
Do you believe pharmacists have a role in educating athletes to avoid unintentional doping?	Yes (94%)No (6%)

WADA = world anti-doping agency; NSAIDs = non-steroidal anti-inflammatory drugs; PPIs = proton pump inhibitors.

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
