# Peer review of "Pharmacists as a Source of Advice on Medication Use for Athletes"

_pharmacy, 2020, doi:10.3390/pharmacy8010010_

Round 1

Reviewer 1 Report

General comment:

This is a brief survey (15 answers requested) among Australian pharmacists about their knowledge of prohibited substances in athletes. Since only 10% of the questionnaires were returned (135 responses), the survey is not really representative, rather preliminary. Therefore the scientific statement of the manuscript is limited. Examples of the list of prohibited drugs for athletes of the World Anti-Doping Agency (WADA) are presented and discussed. The article reminds readers that athletes should not receive medications that are listed as doping agents. The discussion is very short.

Specific comments:

Line 116: The results of the study indicate that upskilling is necessary to prevent unintentional doping. The authors should present some ideas  to upskill pharmacists to expand their knowledge in general or by updating the WADA list annually to be able to advise the athletes adequately.  

The authors should also describe he next efforts that has to be made to qualify pharmacists as specialists for unintentional doping.

Reviewer 2 Report

From a grammatical perspective the paper was well-written. However, the response rate of the survey was low and the data obtained is limited by the survey instrument. For example respondents were asked only a few questions and the range of questions was limited.

Reviewer 3 Report

Overall comments: Medical advice for athletes on use of prohibited medication could perhaps be a potential role for pharmacy. The authors argue that non-elite athletes have to rely on counseling from doctors and pharmacists on how avoid use of unintended prohibited drugs whereas elite athletes usually get help from other sources to avoid using this medication. Yet, the primary research question regards professional athletes whom the authors just concluded do usually not need the help of pharmacist! The aim of the study is thereby very unclear – does it relate to elite or non-elite athletes? Further, I´m unfortunately not convinced by the introduction in the present form that there is a need for pharmacists/ a study in this field. The authors mention one incident of ‘accidental doping’ – but how big is really the problem) In addition, if non-elite athletes use this medication but unintended (meaning for illness purposes) – is there really a problem?

Secondly, these is practically no discussion of the results – why are pharmacists interested but not capable of providing this advice, should this role be on the top of the agenda of what to focus on in the pre- and post-education of pharmacists (should it be provided only for those who don´t watch sports in tv – or how do you intend to use this result going forward) , if yes to include in education how do you propose how pharmacists in practice can approach customers at the pharmacy counter. Further, the methods are not explicated nor are the limits of the study discussed – in particular the consequences of the response rate of 10%.

Specific comments

Abstract: methods: Please describe what the survey intended to explore and which methods of analysis were used.

Paper:

Introduction: Please specify what is the aim of the study and why it is needed – see my comments above.

Materials and methods:

1277 pharmacists were approached – is this a selection or does it cover all pharmacists – if only a selection – please explicate on which grounds this selection was based and the consequences it might have for the results.

Please specify how the survey was developed including the answer categories – for example only being able to answering yes/no to having sufficient knowledge seems quite rough categories – did you run a pilot to make sure questions and answer categories were sufficient? Also the answer categories for frequency of watching sports don´t seem to be exclusive. What was your background for developing the survey? In addition it would perhaps have been relevant to divide athletes in the survey into elite and non-elite – right now the authors don´t know if the respondents answered the question for both groups or only for one of them – thereby it’s not clear whether the primary research question was really answered.

Please explicate the analyses – which methods were used to specify which research questions?

Results

Please provide some demographic data over participants (I assume that more data about them were collected than just their workplace?).

Discussion:

Is practically missing – please see above including discussing strengths and limits of the study.

How can the authors conclude: ‘the results of our survey indicate that there could be misguided trust from both sides’ when you only invited pharmacist to the survey – and I don´t see any questions where the pharmacists were asked about their perceptions of patients’ expectations?

Reviewer 4 Report

Overall this topic of pharmacist involvement in sports medicine and prohibited substances within athletics an interesting area. The topic itself has implications within Australia, but would be very interesting to expand to other areas. It would also be interesting to see this study scaled to a greater number of pharmacists and countries. 

The introduction might be strengthened by inclusion of the role of pharmacists within this specific area and if interacting with athletes is a common practice. In addition, a mention of any curricular or educational requirements or standards might help with clarity. Otherwise, it does have a good balance and citation provided. 

Methods overall are clear and to the point. Can be improved be including additional details. For example, where there any reminders sent out? An explanation of why the publicly available resources were used (why these specifically?) may also help. 

Results are also concise and to the point. Would be helpful to see what "other" primary workplaces the 25% of pharmacists answered are. It also may be helpful to see a breakdown of which demographics based on workplace answered what? For example, did community based pharmacist answer "yes" more often to the question about having sufficient knowledge? 

Discussion can also be expanded on. What other studies or literature are there within this area? Is this the first of its kind? Have other countries or health professionals been surveyed on this topic? How might this apply or be expanded in the future through other studies or countries? What could help to fill in the knowledge gaps that pharmacists are reporting? I think this part of the paper has more potential to improve and expand. 

Conclusion is also brief, but concise. Overall, I believe expanding on the discussion would lead to a little more expansion in the conclusion as well. 

Overall, I believe this is a very timely topic and opportunity for pharmacists across the country and globe. A topic that is widely discussed in other areas, but less in pharmacy and recognition of the role of pharmacists. Great topic! 

Round 2

Reviewer 2 Report

The study is limited by the range, type and number of questions that are asked in the survey tool. As a result, the findings and scope of the study are limited in the present form. Do the authors have additional data that could be presented.

Reviewer 3 Report

Overall comments:

I think the manuscript has been improved by now providing a better rationale for studying the role of pharmacists advising about use of drugs on the doping list for both professionals and non-professionals. However, I think some methodological specifications still need to be provided and I think the potential role of pharmacist in this area also still has to be specified. The discussion is, hence, still too superficial.

Specific comments:

Abstract:

Methods: the analyses have still not been specified.

Introduction:

I think line 150-152 from the Conclusion (‘This is particular important, given the new WADA specifications..) should be moved to the Introduction.

Further, I would like a specification, what level of advice do you think is relevant for pharmacist in this field. What type of knowledge should they have? (knowing that the doping list exist? knowing which type of athletes it covers? knowing which drugs are on the list?) And how do you think this specific knowledge should be used in practice – waiting for pharmacy customers to ask questions or actively asking people of a certain age or running campaigns? These matters should also be addressed in the discussion in relation to the results of the study.

Materials and methods:

How many pharmacists is there in Australia – hence 1277 pharmacist – how big a percentage is this?

How was the survey face-validated?

Please specify the content of the survey – which questions were asked and why.

Results:

Please specify who was the typical participant in the survey in terms of age and gender.

Discussion:

Please explicate which specific role you see for pharmacist in this field in the future (see comments under the introduction) and how your results influence what should now been done. And how big a priority is this as compared to other tasks of the pharmacist?

Which specific studies do you think would be useful going forward?

Limitations: please specify if you think a certain type of pharmacist was represented in the sample and how this influences your results. Likewise describe how the limits you mention influence the results.      
